# Assessing movement quality in individuals with Duchenne muscular dystrophy utilizing accelerometry: Comparisons with healthy controls

Nicholas Joy[1]*, Thomas J. Donnelly[2], Jonathan Soslow[1], William Bryan Burnette[3], Christopher Spurney[4], Nazia Husain[5], Katheryn Gambetta[5], Brian D. Soriano[6], Frank J. Raucci Jr.[7], Kan Hor[8], Larry W. Markham[1], Kimberly Crum[1], Catherine E. Lang[9], Allison E. Miller[9], Jaclyn Tamaroff[10], For the DMDCCC Investigators[¶]

1 Division of Pediatric Cardiology, Department of Pediatrics, Vanderbilt University Medical Center, Nashville, Tennessee, United States of America, 2 University of Tennessee Health Science Center College of Medicine, Memphis, Tennessee, United States of America, 3 Nemours Children's Health, Jacksonville, Florida, United States of America, 4 Children's National Heart Institute, Children's National Hospital, Washington, District of Columbia, United States of America, 5 Division of Cardiology, Department of Pediatrics, Ann & Robert H Lurie Children's Hospital of Chicago, Northwestern University Feinberg School of Medicine, Chicago, Illinois, United States of America, 6 Division of Cardiology, Department of Pediatrics, Seattle Children's Hospital, University of Washington School of Medicine, Seattle, Washington, United States of America, 7 Division of Cardiology, Department of Pediatrics, Children's Hospital of Richmond at VCU, Richmond, Virginia, United States of America, 8 Division of Pediatric Cardiology, Department of Pediatrics, The Ohio State University College of Medicine, Columbus, Ohio, United States of America, 9 Program in Physical Therapy, WashU Medicine, St Louis, Missouri, United States of America, 10 Division of Pediatric Endocrinology and Diabetes, Department of Pediatrics, Vanderbilt University Medical Center, Nashville, Tennessee, United States of America

¶ Membership of DMDCCC Investigators is provided in the Acknowledgements.
☯ These authors contributed equally to this work.
* nicholas.joy@vumc.org

## Abstract

Duchenne muscular dystrophy (DMD) is characterized by progressive decline in skeletal muscle function leading to loss of ambulation and premature cardiopulmonary failure. The ability to monitor declines in skeletal muscle function in a free-living setting would be advantageous. Prior studies have utilized accelerometer measures of movement quantity (e.g., counts per minute, fraction of activity time), but accelerometry research on measures of movement quality in DMD is limited. The aim of the study was to compare quality of movement between a healthy control cohort and individuals with DMD using accelerometry. Accelerometer data were obtained from one study visit for each healthy control (N = 92; ActiGraph Link GT9X, GT3X-BT or a combination) and one to three study visits for each participant with DMD (N = 100; Link GT9X). Measures included counts per minute, entropy, jerk, and movement frequency (mean and standard deviation). Median (IQR) of each measure was reported for each group, including healthy controls and both ambulatory and non-ambulatory DMD participants, and significant differences across each group were compared

the Creative Commons Attribution License, which permits unrestricted use, distribution, and reproduction in any medium, provided the original author and source are credited.

**Data availability statement:** The data for the typically developing children cohort is available on the NICHD DASH repository (DOI: 10.57982/fayx-p832; 10.57982/72z7-m179). The data for the Duchenne muscular dystrophy cohort is available through the Rare Disease Cures Accelerator - Data and Analytics Platform (RDCA-DAP) repository funded by FDA Grant U18FD005320 and administered by Critical Path Institute (C-Path). Both repositories provide free and easy access to anyone of interest.

**Funding:** Data from typically developing children were harmonized and shared using funding by National Institutes of Health grants (R37HD068290 and T32HD007434 to CL). Dr. Raucci is funded by the National Heart, Lung, and Blood Institute (K08HL155852 to FR). Dr. Soslow is supported by the National Heart, Lung, and Blood Institute (K23HL123938, R56HL141248, and R01HL167969 to JS), the Food and Drug Administration (FDA) (1R01FD006649 to JS), and National Center for Advancing Translational Science (UL1-TR002243 to JS). Dr. Soslow has received a grant from Ametris, LLC. to conduct an observational study, however no associated data was used in development of this manuscript. Dr. Tamaroff is supported by the National Institute of Diabetes and Digestive and Kidney Diseases (3R01DK118407-03S1 to JT) and the American Heart Association (23SCEFIA1156470 to JT). This project was supported by Fight DMD. The funders had no role in study design, data collection and analysis, decision to publish, or preparation of the manuscript.

**Competing interests:** I have read the journal's policy and the authors of this manuscript have the following competing interests: JS has received a grant from Ametris, LLC. to conduct an observational study, however no associated data was used in development of this manuscript.

using Mann-Whitney U tests. Correlations were assessed between accelerometer measures of movement quantity and quality, and predictive change in DMD ambulatory status was assessed using longitudinal regression. Significant differences (P<0.01) were observed in all measures between healthy controls, ambulatory DMD, and non-ambulatory DMD participants. Most measures were lower in DMD participants, suggesting decreased movement. Movement frequency values were higher in DMD (Healthy Controls 3.19 [3.05-3.45], Ambulatory DMD 3.60 [3.43-3.89], Non-Ambulatory DMD 4.32 [4.04-4.52]), suggesting more disordered movements. Counts per minute correlated strongly with both jerk (r: 0.722, P<0.05) and mean frequency (r: -0.813, P<0.05). Matched to age, individuals with DMD produce progressively fewer and more disordered (lower quality) movement compared to healthy individuals. Significantly lower entropy and jerk may be explained by a progressive decline in the strength of movements produced by individuals with DMD.

## Author summary

In this study, we evaluated accelerometer-derived measures that have not previously undergone comprehensive study in DMD, including entropy, jerk, and mean frequency. We observed increases in mean frequency among individuals with DMD relative to healthy controls, and these reflect an increase in the randomness of movement as muscle tone progressively decreases. Decreases in jerk and entropy in individuals with DMD compared to healthy individuals reflect decreases in the strength and number of their overall movements. We found strong correlations between movement quantity and quality measures in the DMD cohort as well as strong fit of a model predicting ambulatory status using jerk and mean frequency. Both associations show potential for accelerometry as a tool for upper limb outcomes assessment in both ambulatory and non-ambulatory individuals with DMD. We emphasize the need for further investigation of these key movement quality measures with other clinical assessments to assess longitudinal relationships in individuals with DMD and strengthen the predictive model for loss of ambulation, an important milestone in the progression of DMD.

## 1. Introduction

Duchenne muscular dystrophy (DMD) is an X-linked recessive disorder affecting the dystrophin protein caused by a loss-of-function mutation in the *DMD* gene [1]. The condition has an estimated global incidence of 1 in every 3000–5000 live male births [2]. The progressive disorder is characterized by decline in skeletal and cardiac muscle function leading to loss of ambulation, decreased pulmonary capacity, cardiomyopathy, and premature death—at a median age of 27 (though some variation in life expectancy from the late teens to early 40's) [3]. Although several validated clinical measures of skeletal muscle performance exist for the DMD population, such as the

North Star Ambulatory Assessment (NSAA), quantitative muscle testing (QMT), magnetic resonance imaging of skeletal muscle (skeletal MRI), and Performance of Upper Limb 2.0 (PUL 2.0), these tests are restricted to a clinical setting limiting both their ecological validity to free-living activity as well as geographic access for patients living a significant distance from a DMD care center [4]. In addition, NSAA, QMT, and PUL 2.0 only measure function in a single moment of time and are effort-dependent. This emphasizes the need for a reliable, valid outcome measure of skeletal muscle function to assess treatment efficacy in daily living.

Accelerometry offers a key advantage in its capability of remote monitoring of skeletal muscle function in a free-living environment for both ambulatory and non-ambulatory patients [5–8]. The technology has recently received increased popularity in studies of DMD cohorts as a measure of skeletal muscle function and the stride velocity 95th centile (SV95C) has been approved as an outcome measure for ambulatory patients by the European Medicines Agency [9]. Other accelerometry-derived measures such as counts per minute, total activity counts, and fraction of time spent in moderate-to-vigorous physical activity capture the amount of movement and correlate with other validated clinical DMD measures including the NSAA, QMT, and T2 skeletal muscle MRI data [6–8].

Although declines in total activity counts and counts per minute have been noted in individuals with DMD following loss of ambulation [5,7], a key milestone in the condition's progressive decline of skeletal muscle function, these measures are limited in their assessment; only assessing condition-associated changes in movement quantity rather than quality. Further, there does not currently exist a validated measure of upper extremity movement using accelerometry for individuals with DMD, a gap that currently limits evaluation in non-ambulatory individuals [9]. In populations beyond DMD, including individuals with Parkinson's Disease, additional accelerometer-derived measures such as entropy, jerk, and movement frequency (mean and standard deviation) have been studied in the upper extremity [10–18]. These measures assess the quality of movement—such as movement regularity, smoothness, and order, respectively.

The first measure of interest—entropy—quantifies the time series variation in movement; here entropy is measured specifically in the highest hour of movement to best capture movement regularity [11,15,17]. Higher values of entropy reflect increased disorder in an individual's movement. The second measure of interest—jerk—is the rate of change in acceleration, [10,12–14,16,18]. Jerk is a traditionally utilized measure of movement smoothness, where higher values of jerk reflect decreased movement smoothness and diminished movement control.[12,14,16]. The third and fourth measures of interest- mean frequency and standard deviation of frequency- are derived from the frequencies on the energy spectrum at which an individual moved over the entire recording period [11,13,17]. For people with DMD, higher mean frequencies might represent compensatory movement at distal limb segments due to proximal muscle weakness (e.g., quick flicks or partial gestures with the wrist). Higher standard deviations of movement frequency would reflect more variability in movement control and, in the presence of less overall movement, could also arise from compensatory movement behaviors.

The primary aim of the study was to compare disease-associated differences in accelerometer-derived measures of movement, the output of skeletal muscle function, between a healthy control cohort and individuals with DMD—including entropy, jerk, and movement frequency to evaluate the quality of movement. We hypothesized that movement quality would be worse in individuals with DMD compared to controls, as quantified by higher entropy, jerk, and mean frequency. We anticipated that those with DMD would have more movement irregularity (higher entropy), less movement smoothness (higher jerk), and more disordered movements (higher and more variable frequency) due to skeletal myopathy progression and decreased muscle tone.

There were several secondary study aims. One secondary aim was to analyze associations between measures of movement quantity-- including median acceleration, standard deviation of acceleration, peak acceleration, and counts per minute- and movement quality-- including entropy, jerk, and mean frequency. Based on previous pediatric referent data and data in adults with stroke [19,20], we anticipate that variables within the same construct (quantity or quality) will be strongly correlated while variables between constructs will be moderately correlated. Additional secondary aims were

exploratory and analyzed trends in accelerometer-derived movement measures as related to skeletal myopathy progression in DMD. These included analyzing longitudinal changes in the accelerometer-derived measures as a function of participant ambulatory status, and to assess the predictive ability of accelerometer-derived measures in identifying a wearer's ambulatory status.

## 2. Methods

### 2.1. Ethics statement

All procedures performed in this study involving human participants were in accordance with the ethical standards of the institutional committees and with the 1964 Helsinki Declaration and its later amendments or comparable ethical standards. The study was approved by the Vanderbilt University Medical Center Institutional Review Board (Nashville, TN) and WashU Medicine Institutional Review Board (St Louis, MO). Informed consent was obtained from participants, or, when participants were below the age of 18 years old, a parent or legal guardian with participant assent.

### 2.2. DMD cohort

Participants (N = 100) with DMD were included from a large, multi-site longitudinal natural history study (VUMC IRB: 181942) [7]. Participants from 7 sites—including Vanderbilt University Medical Center, Nationwide Children's Hospital, Riley Children's Hospital, Children's National Medical Center, Lurie Children's Hospital, Seattle Children's Hospital, and Children's Hospital of Richmond at VCU—provided informed consent. For pediatric participants, parent or guardian consent and participant assent were obtained. Participant age, height, weight, and BMI were collected at the time of study visit. Ambulatory status was classified on a binary: ambulatory vs. non-ambulatory based on clinical narrative from site neurologist. For non-ambulatory individuals, heights were estimated using ulnar length. BMI z-scores were calculated using the CDC 2000 growth charts [21].

### 2.3. Healthy Controls cohort

The age- and sex-matched healthy control data (N = 92) were obtained from a harmonized data set [22–25] and an ongoing cohort study of upper limb motor activity using accelerometry conducted at WashU Medicine in St Louis MO [26]. In all studies that contributed data, participants over the age of 18 provided their informed consent to participate, and participants below the age of 18 provided assent along with parent or guardian assent, and all institutional guidelines were followed. Participant age, height, weight, and BMI were collected at the time of study visit. BMI z-scores were calculated using the CDC 2000 growth charts [21]. Participant cohorts were constructed similarly in age and sex. There were 225 healthy control participants who completed between 48–96 hours of accelerometer wear each. All healthy control males between the ages of 7–37 years (n = 92) were included. All healthy control males older than 37 years, all healthy control males younger than 7 years, and all healthy control females were excluded to match the DMD cohort.

### 2.4. Accelerometer wear parameters

In the DMD cohort, participants received one Link GT9X accelerometer at the time of each of three annual study visits, to be worn on dominant wrist for a 7 day and night period, only removing to bathe or swim. The singular wrist-worn accelerometer was chosen for the DMD cohort to improve participant compliance in the long wear period, based on participant feedback in previous studies [27,28]. In the healthy control cohort, participants received two accelerometers at the time of study visit and were instructed to wear the devices on each wrist for 48–96 hours, depending on the study, only removing to bathe or swim. Since the two cohorts differed in their length of wear (48–96 hours for healthy controls compared to 7 days for those with DMD), averages of the wear period for each participant were chosen as the basis of comparison. The accelerometers used for the control cohort were either ActiGraph Link GT9X, ActiGraph GT3X-BT, or a combination

of the two devices, depending on the study they were a participant of [Ametris LLC, Pensacola, FL, USA]. Research has supported compatibility between the devices, particularly when integrated as activity counts, limiting additional variance in the design [29]. Data were collected at 30 Hz. Once devices were returned to the lab, recorded data were downloaded and visually inspected. The data that support the findings of this study are available from the corresponding author upon reasonable request.

## 2.5. Accelerometer processing

As the DMD cohort only measured wear on the participant's dominant wrist, nondominant wear from the healthy control cohort was excluded from analysis. Accelerometer data were processed using custom-written R code (R Core Team, 2013, version 4.4.2). This process entailed extracting three data files from each wrist sensor using ActiLife software (version 7.2.0, Ametris, Pensacola, Florida): a raw 30 Hz file (in gravitational units), a down-sampled 1 Hz and 15 sec data files (sampled in activity counts). The 30 Hz data were band-pass filtered from 0.2-12 Hz to remove acceleration beyond the bands of human activity. Data in the 1 Hz file were filtered using Ametris' proprietary filtering algorithm, with a maximum gain of 0.759 Hz and minimum -6db at 0.212 Hz and 2.148 Hz. Data were then down-sampled from 30 Hz into 1-second epochs for each axis by summing the 30 samples within each second [30,31]. Fifteen second data were likewise down-sampled from 30 Hz but instead to 15-second epochs for each axis. Accelerations in each axis were combined into activity counts, measured as a single vector magnitude using the formula $\sqrt{x^2 + y^2 + z^2}$. A threshold of $\geq 2$ activity counts was used to determine if the upper limb was active for each 1-second epoch [32,33]. Ten variables reflecting the movement of the upper limb in daily life were computed, with some variables computed from the 1 Hz data and others from the 30 Hz data (Table 1). Summaries of their simplest computations are included in Table 1. The variables of movement quality chosen for this analysis were a subset of those analyzed by Miller et al. [25], selected for their novelty in, and clinical significance to, the DMD population. Variables dependent on accelerometer wear across both upper limbs, including jerk asymmetry, were excluded from analysis due to the single upper limb data available in the DMD cohort. The R code used to process the upper limb accelerometry variables is available on https://github.com/keithlohse/HarmonizedAccelData and archived on Zenodo [34].

Movement entropy, representing the repeatability of the movement, was calculated as sample entropy utilizing the *pracma* package in R (v. 4.4.2) for its independence of data length and consistency [35]. Jerk was calculated as the mean difference in sequential activity accounts as a product of sampling frequency. Mean frequency was calculated as an arithmetic weighted mean of the spectral density, while standard deviation of frequency was calculated as a weighted standard deviation of the spectral density. Both movement frequency variables were computed using the *spectrum* function of the *stats* package in R [36].

Daily entropy, jerk, and movement frequency values were then averaged to give the final participant values for comparison. Additional measures chosen for analysis, based on prior data in DMD cohorts, included median acceleration, standard deviation of acceleration, peak acceleration, total activity counts, counts per minute, and counts per day of wear [4,6,7].

## 2.6. Statistical analysis

Statistical software R (v. 4.4.2) was used for all analysis. A threshold of $p < 0.05$ was utilized for statistical significance on all tests. Accelerometer data for the DMD cohorts was restricted to baseline visit wear for comparisons to healthy controls (first and second aims), as longitudinal data was only available for the DMD cohorts. To address the first aim, Kruskal-Wallis tests evaluated differences between the healthy controls and baseline visit DMD cohorts (ambulatory and non-ambulatory) for each accelerometry variable of movement quantity and quality. The false detection rate was employed to adjust for multiple comparisons.

To address the second aim about relationships between measures of movement quantity and movement quality, correlations were computed. As accelerometer measures of movement quantity are already well-documented in DMD

**Table 1. Accelerometer-derived measures.**

| Variable Name | Variable Description | Source |
| --- | --- | --- |
| Total Activity Counts | The sum of all activity counts in the wear period. | 15-Second Epoch |
| Counts Per Minute | The sum of all activity counts divided by the number of minutes the accelerometer was worn. | 15-Second Epoch |
| Counts Per Day | The sum of all activity counts divided by the number of days the accelerometer was worn | 15-Second Epoch |
| Median Acceleration | Median of the accelerations of the dominant upper limb, in activity counts, when the limb was moving (i.e., excluding seconds when accelerations = 0). | 1 Hz |
| Standard Deviation of Acceleration | The standard deviation of the magnitude of accelerations of the dominant upper limb, in activity counts, when the limb was moving. | 1 Hz |
| Peak Acceleration | The highest magnitude of acceleration of the dominant upper limb, in activity counts. | 1 Hz |
| Entropy | The time series variability from accelerations of the dominant upper limb during the hour of highest activity. Higher values correspond to a more random signal. | 1 Hz |
| Jerk | The average jerk of dominant upper limb, measured in g/sec. Higher values correspond to less smooth movement. | 30 Hz |
| Mean Frequency | The weighted mean of component frequencies from the acceleration time series for the dominant upper limb. | 30 Hz |
| Standard Deviation of Frequency | The weighted standard deviation of component frequencies from the acceleration time series for the dominant upper limb. | 30 Hz |

research [4,5,7,8], just one quantity variable was of interest for primary association analysis: counts per minute. Spearman correlations were first computed between four accelerometer measures of interest: Counts per minute, sample entropy, jerk, and mean frequency. Secondarily, a complete spearman correlation matrix was computed between all movement quantity and quality variables analyzed. Lastly, a sub-analysis was conducted taking correlations among just the baseline visit DMD cohort for consistency comparison. P-values were adjusted for multiple correlations using Holm's method. Correlation strength was evaluated according to thresholds of <0.4 as weak, 0.4-0.7 as moderately strong, and >0.7 as strong [37].

To address the exploratory aim evaluating relationships between ambulatory decline and changes in movement quantity and quality in DMD, log-linear mixed effects models were computed for each accelerometer measure. Longitudinal accelerometry data from all three annual study visits among DMD cohort participants were included in this analysis. Each model was constructed with an accelerometer measure as the dependent variable and participant ambulatory status as an independent variable. Between-subjects variance was accounted for in random effects estimation.

To address the exploratory aim about prediction of ambulatory status from accelerometry variables, logistic regression modeling was employed. First a binary logistic regression model was constructed among participants with DMD to differentiate between non-ambulatory (0) and ambulatory (1) statuses. Using model AIC values for predictor selection, the final model chosen included jerk and mean frequency as predictors (logit(P|ambulatory) ~ jerk + mean frequency). Extending this relationship, ordinal logistic regression was then utilized for both the healthy controls cohort and DMD cohort to attempt to differentiate participant data across three classes: healthy controls, ambulatory DMD, and non-ambulatory DMD using the same accelerometer predictors as the binary model. Each model employed a cross-sectional design, analyzing single study visit data for the healthy controls cohort against baseline study visit data for the DMD cohort. To test the discriminating strength of each model, bootstrapping procedures were employed using the *validate* function of the *rms* package (v. 8.0-0) in R [38]. Each function was resampled 1000 times. Additionally, a small cohort (n = 24) (VUMC IRB: 220381, 231407, 231636) of participants with DMD matched to the age range of the original participants was utilized as an external test for the prediction ability of the binary model.

# 3. Results

## 3.1. Participant demographics

Accelerometer data from 100 male participants with DMD and 92 healthy male control participants were analyzed. Their demographic and anthropometric information is summarized in Table 2. The cohorts did not differ significantly in terms of participant age, weight, race and ethnicity distributions (Table 2). Participant height and subsequent BMI did differ significantly (Table 2) with individuals with DMD being shorter and having higher BMI Z-scores on average.

## 3.2. Disease-associated differences in accelerometer-derived measures

Significant differences were found in all (10/10) accelerometer-derived measures between the healthy controls (n = 92), ambulatory participants with DMD (n = 47), and non-ambulatory participants with DMD (n = 53; Table 3). Adjustment by false detection rate upheld significance of all parameters. Non-ambulatory individuals with DMD had the highest mean frequency and standard deviation of frequency, followed by ambulatory individuals with DMD, then healthy controls. For all other measurements, both ambulatory and non-ambulatory DMD participants resulted in significantly lower values compared to the healthy controls, with the lowest values in those who were non-ambulatory.

## 3.3. Correlations between accelerometry-derived quantity and quality measures

In the primary analysis of associations between quantity and quality variables, four measures—counts per minute, sample entropy, jerk, and mean frequency, were tested for significant correlations between the parameters (Table 4; Fig 1). All correlations were significant, with the strongest between Counts per minute and jerk, and counts per minute and mean frequency (Table 4).

**Table 2. Participant characteristics (N = 192).**

| Participants Characteristics (N = 192) | | | |
|---|---|---|---|
|  | Healthy Controls (N = 92) Median (IQR) or n(%) | DMD Cohort (N = 100) Median (IQR) or n(%) | P-Value |
| Age (years) | 11 (9, 15) | 12 (11, 15) | 0.057 |
| Height (cm) | 147.0 (135.0, 165.5) | 142.2 (131.6, 155.5) | **0.025** |
| Weight (kg) | 43.75 (30.25, 63.45) | 45.20 (33.70, 58.80) | 0.735 |
| BMI z-score | 0.23 (-0.36, 0.85) | 0.90 (0.04, 2.20) | **0.009** |
| Ambulatory (%) | 92 (100%) | 47 (47%) |  |
| **Race** |  |  | 0.752 |
| White | 81 (88.1%) | 78 (78.0%) |  |
| African American | 4 (4.3%) | 7 (7.0%) |  |
| Asian | 5 (5.4%) | 6 (6.0%) |  |
| Multiracial | 2 (2.2%) | 3 (3.0%) |  |
| Unknown | 0 | 6 (6.0%) |  |
| **Ethnicity** |  |  |  |
| Hispanic | 5 (5.4%) | 10 (10%) |  |

a. Comparison of baseline demographics for the healthy control cohort and DMD cohort. Demographics presented in the format Median (IQR). Racial and ethnic group information presented as N(%).

**Table 3. Comparison of accelerometry-derived measurements in healthy controls, ambulatory, and non-ambulatory individuals with DMD.**

| Accelerometry Measure | Healthy Controls Median (IQR) (n = 92) | Ambulatory DMD Median (IQR) (n = 47) | Non-Ambulatory DMD Median (IQR) (n = 53) | P-Value of Effect |
|---|---|---|---|---|
| Median Acceleration | 78.26 (71.34, 86.31) | 68.33 (55.00, 74.00) | 40.80 (30.50, 49.80) | <0.001 |
| Standard Deviation of Acceleration | 115.95 (95.98, 128.45) | 84.67 (79.00, 92.00) | 59.33 (46.25, 68.75) | <0.001 |
| Peak Acceleration | 1254.82 (1076.33, 1333.09) | 827.80 (682.20, 964.75) | 478.33 (412.40, 592.25) | <0.001 |
| Entropy | 0.73 (0.57, 0.92) | 0.61 (0.41, 0.76) | 0.40 (0.29, 0.57) | 0.016/ <0.001 |
| Jerk | 1.46 (1.01, 2.02) | 0.81 (0.62, 0.96) | 0.41 (0.31, 0.59) | <0.001 |
| Mean Frequency | 3.19 (3.05, 3.45) | 3.60 (3.43, 3.89) | 4.32 (4.04, 4.52) | <0.001 |
| Standard Deviation of Frequency | 2.62 (2.51, 2.70) | 2.69 (2.58, 2.78) | 2.78 (2.60, 2.82) | <0.001 |
| Total Counts | $1.7 \times 10^8$ ($1.3 \times 10^8$, $2.6 \times 10^8$) | $1.5 \times 10^7$ ($1.1 \times 10^7$, $2.0 \times 10^7$) | $0.75 \times 10^7$ ($0.33 \times 10^7$, $1.1 \times 10^7$) | 0.006/ <0.001 |
| Counts Per Minute | 4695.44 (4280.22, 5178.56) | 1643.58 (1275.75, 2014.50) | 852.53 (438.90, 1189.51) | <0.001 |
| Counts Per Day | $6.7 \times 10^6$ ($6.1 \times 10^6$, $7.5 \times 10^6$) | $2.3 \times 10^6$ ($1.8 \times 10^6$, $2.9 \times 10^6$) | $1.2 \times 10^6$ ($6.1 \times 10^5$, $1.7 \times 10^6$) | <0.001 |

[a.]P-values for significance of median difference between healthy controls and ambulatory versus non-ambulatory DMD participants. Descriptive statistics of accelerometer measures presented in the format Median (IQR).

**Table 4. Correlations of accelerometry-derived measures across all cohorts.**

| Measure | Counts Per Min | Entropy | Jerk | Mean Frequency |
|---|---|---|---|---|
| Counts Per Min | – | 0.674* | 0.898* | -0.813* |
| Entropy | 0.674* | – | 0.722* | -0.487* |
| Jerk | 0.898* | 0.722* | – | -0.719* |
| Mean Frequency | -0.813* | -0.487* | -0.719* | – |

[a.]Significant coefficients indicated by *.

### 3.4. Decline in accelerometer-derived measures as function of ambulatory status

Log-linear mixed effects modeling showed highly significant effects of ambulatory status on differences in each accelerometer-derived measure (Table 5). In line with the group comparisons in Table 2, lower acceleration, entropy, and jerk with increases in mean and variability of frequency were associated with loss of ambulation.

### 3.5. Prediction of ambulatory status from accelerometry

Variation in accelerometer quantity and quality measures between cohorts was observed (Fig 1). Binary logistic regression was used to create a model to distinguish ambulatory status among participants with DMD based on the same predictors jerk and mean frequency. A 1 unit increase in participant jerk was found to increase the odds ratio of a classification of ambulatory by a factor of 9.85, holding mean frequency constant. A 1 unit increase in mean

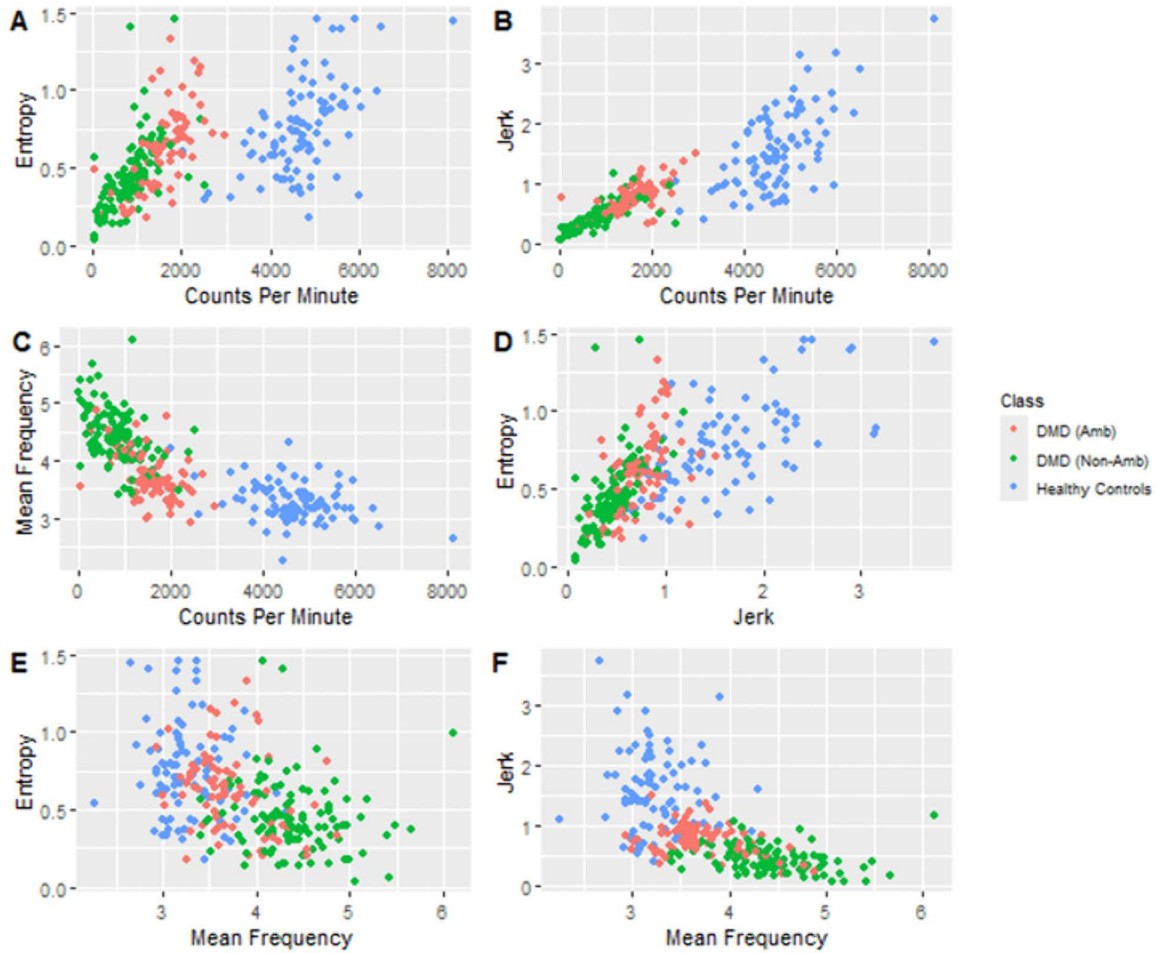

**Fig 1. Variation in accelerometry-derived measures among the control and DMD cohorts.** Plots A-F model relationships between pairs of movement quantity and quality measures. Blue dots represent healthy control datapoints; orange dots represent ambulatory DMD datapoints; green dots represent non-ambulatory DMD datapoints.

frequency was found to increase the odds ratio for a classification of non-ambulatory by a factor of 41274.49, holding jerk constant. The model had an AUC of 0.897. The predicted participant ambulatory class was accurate among 83.6% of wear periods tested (Fig 2). Bootstrapping demonstrated capability in the model discriminating ambulatory statuses. The marginal effect of participant age was found to be non-significant in the logistic model. Similarly, when tested against the external DMD cohort (n = 24), the binary model accurately predicted 79.2% of the participants' ambulatory statuses. Among participants who received an incorrect prediction, their true ambulatory classification ranged from late ambulatory to early non-ambulatory stage. The model was visualized with a 3-dimensional scatter plot (Fig 2).

Ordinal logistic regression was used to create a model to distinguish between healthy controls, ambulatory DMD participants, and non-ambulatory DMD participants. The model was weak in classifying non-ambulatory participants with DMD, with AUC 0.688. However, the model was able to distinguish between healthy control and DMD (either ambulatory or non-ambulatory).

PLOS Digital Health

**Table 5. Longitudinal relationship of ambulatory status and accelerometry-derived measures among DMD participants (N = 100 Participants, n = 183 wear periods).**

| Accelerometry Measure | Fixed Effects B | Random Effects Intercept $s^2$ | Log-Likelihood |
|---|---|---|---|
| Median Acceleration | -0.49* | 0.03 | -32.01 |
| Standard Deviation of Acceleration | -0.41* | 0.01 | 16.08 |
| Peak Acceleration | -0.45* | 0.02 | -0.64 |
| Entropy | -0.36* | 0.12 | -125.52 |
| Jerk | -0.53* | 0.07 | -102.77 |
| Mean Frequency | 0.15* | 0.01 | 152.09 |
| Standard Deviation of Frequency | 0.03* | 0.00 | 234.28 |
| Total Counts | -0.70* | 0.26 | -227.78 |
| Counts Per Minute | -0.51* | 0.13 | -178.10 |
| Counts Per Day | -0.53* | 0.14 | -185.56 |

[a.]Estimates of fixed-effects slope coefficient, random effects variance associated with participant, and log-likelihood of linear model. A positive B corresponds to a higher accelerometer measure in non-ambulatory DMD participants while a negative B to a lower estimate. Higher log-likelihood corresponds to greater model fit. Significance indicated by *.

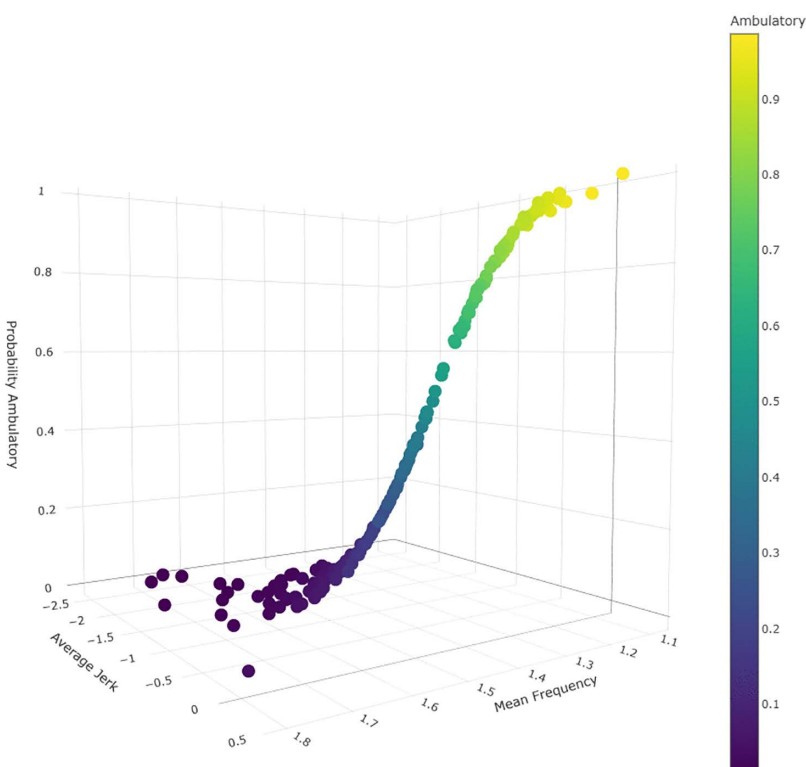

**Fig 2. Predicted DMD participant ambulatory status using accelerometer parameters jerk and mean frequency.** Probability of classification as ambulatory visualized by color gradient.

## 4. Discussion

Due to progressive skeletal myopathy associated with DMD, we expected that individuals with DMD would produce movements of higher entropy, jerk, and mean frequency. In comparison with healthy individuals, however, individuals with DMD produced on average lower entropy and jerk, and higher mean frequency (Table 3). In addition, accelerometer-derived measurements of movement quantity and quality were associated with moderate strength among both participants with DMD and healthy controls. Taken together, accelerometer-derived measures of both movement quality and quantity show potential as a remote, free-living assessment of the skeletal muscle function for individuals with DMD.

The median entropy and jerk of movements in participants with DMD were lower when compared to healthy controls (Table 3). This was contrary to our expectations. In our hypothesis, we assumed that as a function of muscle loss, individuals' movements would become less regular (higher entropy) and more clumsy (higher jerk). Instead, we found lower entropy and lower jerk in those with DMD. We postulate, however, that the lower entropy and jerk in those with DMD is a result of progressive decline in the strength of movements produced by individuals with DMD, resulting in less overall movement. This hypothesis is strengthened by the pattern of entropy and jerk from healthy controls (highest) to ambulatory individuals with DMD (middle) to non-ambulatory individuals with DMD (lowest). Our result is consistent with another study [39] which measured cumulative jerk in a small cohort of non-ambulatory individuals with DMD. There, jerk was positively correlated with elbow flexion strength indicating those with less severe muscular effect (increased elbow flexion strength or ambulatory) have higher jerk.

The hypothesis of an increase in mean frequency in those with DMD, however, was supported by the findings (Table 3). A higher mean frequency of movement supports increased disorder in movement and could arise from the DMD-associated skeletal myopathy in more proximal muscles. There was also a small but significant difference in the standard deviation of frequency between healthy controls and participants with DMD. This small increase likely reflects increasing randomness of movement associated with progression in DMD.

Correlations between measures of movement quantity and quality among participants with and without DMD were moderately strong and significant (Table 4). These correlations are in line with previously published referent data [17]. Multiple previous accelerometry studies in DMD cohorts have utilized counts per minute as a key outcomes metric of skeletal muscle activity, and its correlations with key clinical measures of skeletal muscle strength such as quantitative muscle testing (QMT), and the North Star Ambulatory Assessment (NSAA) demonstrated [4,5,7]. The strong correlations drawn between both jerk and mean frequency and counts per minute in this study (Table 4; Fig 1) raise the question of how strongly these measures would correlate with other clinical outcomes assessment data in DMD patients. Unfortunately, NSAA was not available for this cohort as a significant number of DMD patients were non-ambulatory. The inability to evaluate NSAA in non-ambulatory participants highlights the importance of investigating accelerometry metrics that can be used in both ambulatory and non-ambulatory individuals. Additional correlations computed between acceleration parameters and counts per minute also demonstrate strong convergent validity, with significant correlation strengths as high as 0.9 in magnitude, indicating substantial shared variance between variables. Thus, one might consider that any one of these variables could be deployed to index movement behavior in the DMD population.

A binary model including mean frequency and jerk was able to discriminate between ambulatory and non-ambulatory patients with DMD (Fig 2). This suggests that accelerometry measures not only correlate with progression of disease, as previously described, they also associate strongly with important clinical outcomes of interest (i.e., loss of ambulation). This linking of outcome measures to a life-altering event is a primary focus of the Food and Drug Administration when evaluating outcome measures [40]. As the training dataset only consisted of 183 wear periods, while the external cohort test consisted of just 24, the 83.6% prediction accuracy of the binary regression model may actually underestimate or overestimate its true accuracy, and a larger future cohort is needed to strengthen these findings. Additionally, the progressive nature of DMD and loss of ambulation could support the determination of a particular range, rather than binary thresholds, for the jerk and mean frequency measures to predict loss of ambulation. Loss of ambulation in DMD is a

spectrum (from late ambulatory to early non-ambulatory to late non-ambulatory) with progressive inability to ambulate over longer and more complex distances over time. All wear periods misclassified by the regression model were within 2 years (before or after) of the participant's loss of ambulation. While the individual may still retain the ability to walk short distances during this period, often the lead up to loss of ambulation will include increased utilization of mobility assistive technology. Expanding prediction to a range rather than a strict cutoff could improve accelerometry's anticipation of loss of ambulation for families, which is particularly of interest in the use of accelerometry remotely.

Interestingly, the ordinal logistic regression model comparing all three classes was able to distinguish between healthy patients and DMD but had poorer performance distinguishing between ambulatory and non-ambulatory participants with DMD. This may have been partly attributed to the large magnitude difference in skeletal muscle function between healthy control participants and DMD participants.

There are many outcome measures that have been used in patients with DMD. Unfortunately, the majority of these are effort-dependent and/or require ambulation. While skeletal muscle MRI is objective, effort independent and agnostic to ambulatory status, the FDA considers MRI a surrogate outcome measure, limiting its use as a clinical trial endpoint. Upper limb accelerometry measures are effort-independent, free-living, and direct measures of a patient's skeletal muscle function, fulfilling the FDA's requirements as an outcome measure. It is critical to identify the most sensitive and reproducible accelerometry metrics. In this manuscript, we evaluated the potential candidacy of multiple measures. These measures can also be leveraged in clinical care, providing data on skeletal muscle function over an extended period, as opposed to a single point in time for most current clinical measures.

Future studies should compare the accelerometer-derived measures of entropy, jerk, and movement frequency in both ambulatory and non-ambulatory individuals with DMD with other clinical outcome assessments such as cardiac MRI, QMT, and pulmonary function testing (PFT) data, to serve as a reference point of skeletal and pulmonary function. A comparison of these measures with other accelerometer-derived measures such six-minute activity 95th centile (6M95c) would also be beneficial [7]. How these measures are affected by participant cardiac function, and new DMD medications and gene therapies is of interest. Additionally, while all-day wear was analyzed in both cohorts, sub analysis of measure differences between movements while awake versus asleep was not conducted. Lastly, an analysis of longitudinal changes in these quality measures in individuals with DMD, particularly surrounding the major milestone of loss of ambulation, would be of benefit.

## 5. Limitations

Four key limitations need to be considered in the interpretation of these data and consideration of future investigation. First, this study classified ambulatory status as a binary outcome determined by the treating the neurologist. Therefore, changes in the accelerometer measures across different levels within these two statuses could not be analyzed. Future analysis incorporating a more stratified ambulatory status classification would be of clinical utility. Second, the cohorts were collected separately, as part of different studies with different protocols. Despite this, the demographics between cohorts are very similar (Table 1), minimizing confounding factors and facilitating identification of the effects of DMD-associated skeletal myopathy progression on daily movement. We saw an expected difference between cohorts in the anthropometric values (height, BMI); shorter stature and increased BMI are common complications of DMD [41]. It is unlikely that anthropometric differences caused the differences between groups, because distributions of jerk in typically developing infants' unstructured movements show substantial overlap (1.41 [1.19, 1.67] g/s) [42] with the typically developing children included in this study (1.46 [1.01, 2.02] g/s) (Table 3). Third, a limitation to the accelerometer measure in this study, particularly for non-ambulatory participants, is the lack of an available adjustment for the acceleration of wheelchair or other assistive movement device, along with the limited clinical data presented on compensatory upper limb movements. Non-ambulatory patients may be utilizing power-wheelchairs or manual wheelchairs which involve different upper limb motions and differing amounts of potential noise in the accelerometer's capture of upper extremity movement. Future work is needed to provide an algorithm that adjusts for this additional movement captured by the accelerometer sensor in order

that wearers using such assistive mobility devices receive truly representative evaluation of their skeletal muscle function. The utility of upper limb assessment data, such as from the PUL 2.0 in accelerometer analysis has previously been suggested [7]. Clinical assessments of upper limb movement such as the PUL 2.0 could provide an additional referent measure of compensatory upper limb movements detected in accelerometer measure and contribute meaningful information to developing such an algorithm adjusting for the assistive mobility devices. Fourth, a limitation in the study design, particularly affecting the DMD cohort, was the duration of time between assessments. As accelerometer data was only captured annually in a 7-day period, modeling the true longitudinal progression of skeletal muscle performance is challenging. In future clinical trials, the frequency and duration of accelerometer wear could be altered (e.g., every 3–4 months instead of annually, 3–4 days at a time instead of 7 days). Increasing the frequency of data capture would increase the number of time points to analyze gradual changes in muscle performance, while decreasing the duration of the wear period could also lessen participant burden with the device wear, indirectly improving compliance with the device as well.

## Acknowledgments

We thank the children and families who participated in this study. We would also like to thank Keith Lohse, PhD for his efforts developing the code for the accelerometer measures. Membership of DMDCCC Investigators is as follows: Jonathan Dayan, Division of Pediatric Cardiology, Department of Pediatrics, University of California Davis; M. Jay Campbell and Jennifer Li, Division of Pediatric Cardiology, Department of Pediatrics, Duke University Medical Center; Beth Kaufman, Division of Pediatric Cardiology, Department of Pediatrics, Lucile Packard Children's Hospital at Stanford; Carol Wittlieb-Weber, Division of Pediatric Cardiology, Department of Pediatrics, Children's Hospital of Philadelphia; Teresa Wang, Division of Cardiovascular Medicine, Department of Medicine, University of Pennsylvania

The content is solely the responsibility of the authors and does not necessarily represent the official views of the National Institutes of Health.

## Author contributions

**Conceptualization:** Catherine E. Lang, Allison E. Miller, Jaclyn Tamaroff.

**Data curation:** Nicholas Joy, Thomas J. Donnelly, Jonathan Soslow, Christopher Spurney, Kimberly Crum, Catherine E. Lang, Allison E. Miller.

**Formal analysis:** Nicholas Joy, Jonathan Soslow, Allison E. Miller.

**Investigation:** Nicholas Joy, Jonathan Soslow, William Bryan Burnette, Nazia Husain, Katheryn Gambetta, Brian D. Soriano, Frank J. Raucci Jr., Kan Hor, Larry W. Markham, Catherine E. Lang.

**Methodology:** Nicholas Joy, Jonathan Soslow, Kimberly Crum, Catherine E. Lang, Allison E. Miller, Jaclyn Tamaroff.

**Project administration:** Jonathan Soslow, Kimberly Crum, Catherine E. Lang.

**Resources:** Jonathan Soslow, Kimberly Crum, Catherine E. Lang.

**Software:** Nicholas Joy, Thomas J. Donnelly.

**Supervision:** Jonathan Soslow, Catherine E. Lang, Jaclyn Tamaroff.

**Validation:** Nicholas Joy, Catherine E. Lang, Allison E. Miller.

**Visualization:** Nicholas Joy.

**Writing – original draft:** Nicholas Joy, Jonathan Soslow, Catherine E. Lang, Jaclyn Tamaroff.

**Writing – review & editing:** Nicholas Joy, Thomas J. Donnelly, Jonathan Soslow, William Bryan Burnette, Christopher Spurney, Nazia Husain, Katheryn Gambetta, Brian D. Soriano, Frank J. Raucci Jr., Kan Hor, Larry W. Markham, Kimberly Crum, Catherine E. Lang, Allison E. Miller, Jaclyn Tamaroff.

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
