## [Decision Letter · Decision Letter 0]

14 Apr 2026

PDIG-D-26-00377Assessing movement quality in individuals with Duchenne muscular dystrophy utilizing accelerometry: Comparisons with healthy controlsPLOS Digital HealthDear Dr. Joy,Thank you for submitting your manuscript to PLOS Digital Health. After careful consideration, we feel that it has merit but does not fully meet PLOS Digital Health's publication criteria as it currently stands. Therefore, we invite you to submit a revised version of the manuscript that addresses the points raised during the review process.Please submit your revised manuscript by May 14 2026 11:59PM. If you will need more time than this to complete your revisions, please reply to this message or contact the journal office at digitalhealth@plos.org. Please include the following items when submitting your revised manuscript:* A letter that responds to each point raised by the editor and reviewer(s). You should upload this letter as a separate file labeled 'Response to Reviewers'. This file does not need to include responses to any formatting updates and technical items listed in the 'Journal Requirements' section below.* A marked-up copy of your manuscript that highlights changes made to the original version. You should upload this as a separate file labeled 'Revised Manuscript with Track Changes'.* An unmarked version of your revised paper without tracked changes. You should upload this as a separate file labeled 'Manuscript'.If you would like to make changes to your financial disclosure, competing interests statement, or data availability statement, please make these updates within the submission form at the time of resubmission. Guidelines for resubmitting your figure files are available below the reviewer comments at the end of this letter.We look forward to receiving your revised manuscript.Kind regards,Lawrence HayesSection EditorPLOS Digital HealthLeo Anthony CeliEditor-in-ChiefPLOS Digital Healthorcid.org/0000-0001-6712-6626**Journal Requirements:**

i. Please clarify all sources of financial support for your study. List the grants, grant numbers, and organizations that funded your study, including funding received from your institution. Please note that suppliers of material support, including research materials, should be recognized in the Acknowledgements section rather than in the Financial Disclosure.

ii. State the initials, alongside each funding source, of each author to receive each grant. For example: "This work was supported by the National Institutes of Health (####### to AM; ###### to CJ) and the National Science Foundation (###### to AM)."

iii. State what role the funders took in the study. If the funders had no role in your study, please state: “The funders had no role in study design, data collection and analysis, decision to publish, or preparation of the manuscript.”

iv. If any authors received a salary from any of your funders, please state which authors and which funders.

2. Please send a completed 'Competing Interests' statement, including any COIs declared by your co-authors. If you have no competing interests to declare, please state "The authors have declared that no competing interests exist". Otherwise please declare all competing interests beginning with the statement "I have read the journal's policy and the authors of this manuscript have the following competing interests:"

3. In the online submission form, you indicated that “The data for the typically developing children cohort is available on the NICHD DASH repository. All data is available upon reasonable request to the author.”.

3. Uploaded as supplementary information.

4. Please ensure that your Ethics Statement is available in its entirety at the beginning of your Methods section, under a subheading 'Ethics Statement'. It must include:

i) The name(s) of the Institutional Review Board(s) or Ethics Committee(s)

ii) The approval number(s), or a statement that approval was granted by the named board(s)

iii) (for human participants or donors) - A statement that formal consent was obtained (must state whether verbal/written) OR the reason consent was not obtained (e.g., anonymity).

vi) (for child participants) - Please update your Ethics Statement with confirmation that written informed consent was obtained from the parent/guardian of each participant under 18 years of age.

5. Please provide separate figure files in .tif or .eps format.

If the reviewer comments include a recommendation to cite specific previously published works, please review and evaluate these publications to determine whether they are relevant and should be cited. There is no requirement to cite these works unless the editor has indicated otherwise. **Additional Editor Comments (if provided):****Reviewers' Comments:**Reviewer's Responses to Questions

**Comments to the Author**

1. Does this manuscript meet PLOS Digital Health’s publication criteria? Is the manuscript technically sound, and do the data support the conclusions? The manuscript must describe methodologically and ethically rigorous research with conclusions that are appropriately drawn based on the data presented.

Reviewer #1: Yes

Reviewer #2: Yes

2. Has the statistical analysis been performed appropriately and rigorously?

Reviewer #1: Yes

Reviewer #2: Yes

3. Have the authors made all data underlying the findings in their manuscript fully available (please refer to the Data Availability Statement at the start of the manuscript PDF file)?

Reviewer #1: Yes

Reviewer #2: Yes

4. Is the manuscript presented in an intelligible fashion and written in standard English?

Reviewer #1: Yes

Reviewer #2: No

5. Review Comments to the Author

**Reviewer #1:** Dear Author,

Thank you for the nice work titled “Assessing movement quality in individuals with Duchenne muscular dystrophy utilizing accelerometry: Comparisons with healthy controls”. My comments on this study, which was conducted to perform movement analysis in patients with DMD using accelerometry and compare the results with healthy individuals, are as follows;

I believe the subject of this study as evaluating the usability of a promising assessment method in terms of both accessibility and cost-effectiveness, for predicting ambulatory status of children, by evaluating not only the amount but also the quality of movement in their natural living environments without being dependent on a clinic is both an exciting topic and an important step towards filling a gap in the literature, as it points to a method that can be used for DMD patients at all stages and has the potential to be used in monitoring loss of ambulation. Ultimately, your results indicating the possible effective usability of accelerometry for this purpose and its ability to differentiate between ambuatory and non-ambulatory individuals as well as the healthy and DMD subjects, helps to shed light on an important issue in the literature in recent years. I congratulate the researchers for this study, which will make a significant contribution to the field. I believe the work is acceptable for publication, provided the following minor corrections are made.

1. In the Abstract, it is recommended to provide sufficient information about the research methodology, including which descriptive data were recorded for each group, which accelerometer was used, the data collection process, etc.

2. In the Abstract, it is recommended to present the research results with numerical data, including the descriptors of the groups and comparison results with healthy individuals.

3. In the Method section, the distribution in Table 3 could be corrected by providing separate data for all ethnic groups.

4. In the Methods section, was ambulatory status recorded only as walker/non-walker? Or was it assessed according to any other classification? If applicable, a comparison according to ambulatory level could also be added, so that information on which parameters of the quantity and quality of movement changed at different ambulatory levels would provide valuable information to rehabilitation specialists.

**Reviewer #2:** I appreciate the opportunity to review the manuscript.

I congratulate the authors on the topic presented, and on the number of participants, considering it is a rare disease.

The manuscript presents a scientifically and clinically relevant topic for the population with DMD.

The introduction provides an overview of what is known about the topic and indicates existing gaps.

The method is adequate for the study's purpose.

It would be interesting to specify the participants' Signs, since in ambulatory individuals, the Signs can range from 1 to 6, and in non-ambulatory individuals from 7 to 10. This is a point to be included in the characterization of the sample and to verify possible relationships, since this scale indicates the staging of the disease - the level of functionality of the individual with DMD.

Regarding the results, I suggest reviewing the formatting of all tables, as they are presented as charts. Tables have open sides.

Regarding tables 3 and 5, include the mean and standard deviation of the evaluated variables on the same line, for example: mean acceleration and standard deviation of acceleration.

In the first part of table 2 and in table 3, indicate what the values in parentheses mean. I believe they are the minimum and maximum values of the studied variables.

I suggest adding the clinical applicability of the study findings to the discussion. This needs to be explored

The discussion will present, as a limitation, the importance of assessing upper limb function in ambulatory and non-ambulatory individuals using clinical scales that indicate upper limb performance and compensations, as well as the compensatory strategies used by this population to maintain function, and how this could be explored in future studies.

6. PLOS authors have the option to publish the peer review history of their article (what does this mean?). If published, this will include your full peer review and any attached files.

**Do you want your identity to be public for this peer review?** For information about this choice, including consent withdrawal, please see our Privacy Policy.

Reviewer #1: No

Reviewer #2: No

**Figure resubmission:**While revising your submission, we strongly recommend that you use PLOS’s NAAS tool (https://ngplosjournals.pagemajik.ai/artanalysis) to test your figure files. NAAS can convert your figure files to the TIFF file type and meet basic requirements (such as print size, resolution), or provide you with a report on issues that do not meet our requirements and that NAAS cannot fix.
---

## [Editor Report · Decision Letter 1]

12 May 2026

Assessing movement quality in individuals with Duchenne muscular dystrophy utilizing accelerometry: Comparisons with healthy controls

PDIG-D-26-00377R1

Dear Mr. Joy,

We are pleased to inform you that your manuscript 'Assessing movement quality in individuals with Duchenne muscular dystrophy utilizing accelerometry: Comparisons with healthy controls' has been provisionally accepted for publication in PLOS Digital Health.

Best regards,

Lawrence D Hayes, PhD

Section Editor

PLOS Digital Health